# The Process of Plaque Rupture: The Role of Vasa Vasorum and Medial Smooth Muscle Contraction Monitored by the Cardio-Ankle Vascular Index

**DOI:** 10.3390/jcm12237436

**Published:** 2023-11-30

**Authors:** Kohji Shirai, Takashi Hitsumoto, Shuji Sato, Mao Takahashi, Atsuhito Saiki, Daiji Nagayama, Masahiro Ohira, Akira Takahara, Kazuhiro Shimizu

**Affiliations:** 1Research Center, Seijinkai, Mihama Hospital, 1-1-5 Utase, Mihama-ku, Chiba-shi, Chiba 261-0013, Japan; 2Department of Internal Medicine, Toho University Sakura Medical Center, 564-1 Shimoshizu, Sakura-shi, Chiba 285-8741, Japan; shuuji.satou@med.toho-u.ac.jp (S.S.); takahashi-04@sakura.med.toho-u.ac.jp (M.T.); k432@sakura.med.toho-u.ac.jp (K.S.); 3Hitsumoto Clinic, 2-7-7 Takezaki-cho, Shimonoseki-shi, Yamaguchi 750-0025, Japan; thitsu@jcom.home.ne.jp; 4Nagayama Clinic, 2-12-22, Tenjin-cho, Oyama-shi, Tochigi 323-0032, Japan; deverlast96071@gmail.com; 5Department of Diabetes, Metabolism and Endocrinology, Toho University Ohashi Medical Center, 2-22-36 Ohashi, Meguro-ku, Tokyo 153-8515, Japan; 600137om@sakura.med.toho-u.ac.jp; 6Department of Pharmacology and Therapeutics, Faculty of Pharmaceutical Sciences, Toho University, 2-2-1 Miyama, Funabashi-shi, Chiba 274-8510, Japan; akirat@phar.toho-u.ac.jp

**Keywords:** cholesterol oxidative products, atheromatous lesion, plaque rupture, vasa vasorum, cardio-ankle vascular index

## Abstract

A warning sign for impending cardiovascular events is not fully established. In the process of plaque rupture, the formation of vulnerable plaque is important, and oxidized cholesterols play an important role in its progression. Furthermore, the significance of vasa vasorum penetrating the medial smooth muscle layer and being rich in atheromatous lesions should be noted. The cardio-ankle vascular index (CAVI) is a new arterial stiffness index of the arterial tree from the origin of the aorta to the ankle. The CAVI reflects functional stiffness, in addition to structural stiffness. The rapid rise in the CAVI means medial smooth muscle cell contraction and strangling vasa vasorum. A rapid rise in the CAVI in people after a big earthquake, following a high frequency of cardiovascular events has been reported. There are several cases that showed a rapid rise in the CAVI a few weeks or months before suffering cardiovascular events. To explain these sequences of events, we proposed a hypothesis: a rapid rise in the CAVI means medial smooth muscle contraction, strangling vasa vasorum, leading to ischemia and the necrosis of vulnerable plaque, and then the plaque ruptures. In individuals having a high CAVI, further rapid rise in the CAVI might be a warning sign for impending cardiovascular events. In such cases, treatments to decrease the CAVI better be taken soon.

## 1. Introduction

Arteriosclerotic disease is the major cause of mortality and morbidity among cardiovascular diseases not only in developed countries but also in developing countries [1]. Regarding the formation of atherosclerotic lesions in the arterial wall, many hypotheses have been proposed. Cholesterol was first thought to be a cause of atheroma formation [2]. However, cholesterol is an important lipid component in the cell membrane. Then, denatured low-density lipoproteins (LDLs) containing cholesterol were proposed as a cause of form cell formation in atheroma [3,4,5]. Later, Ross et al. proposed an injury response hypothesis [6] to explain intimal thickening composed of the synthetic type of smooth muscle cell (SMC) proliferation, leading to stenosis of the arterial cavity. However, stenosis was not necessarily observed in the occluded coronary artery by thrombus [7]. Then, plaque rupture theory was proposed [8]. In that theory, vulnerable plaque is formed subsequent to a chronic inflammatory reaction in the intima. In chronic inflammatory reactions, macrophages digest the surrounding tissues, thinning a covering cap. When a thin cap ruptures, a formed thrombus occludes the lumen of the artery, leading to myocardial infarction. However, the real causes of provoked inflammation in atheromatous lesions were not fully clarified, although many cytokines relating to inflammatory reactions were investigated. Oxysterol products are toxic to cells [9]. Oxysterol products might be a ringleader to make vulnerable plaque. 

On the other side, it is reported that atheromatous lesions are rich in vasa vasorum [10,11]. The ischemia of intimal atheromatous lesions produces angiogenic factors, such as vascular endothelial growth factor; subsequently, neovascularization develops from the adventitia into intimal atheroma through the medial smooth muscle layer to form a network of vasa vasorum. The significance of vasa vasorum penetrating through the medial smooth muscle layer into intimal plaque has not been fully discussed. 

Arterial stiffness is composed of structural stiffness and functional stiffness. Pulse wave velocity (PWV) has been used as an index reflecting arterial stiffness [12]. However, PWV is essentially affected by blood pressure at the measuring time [13]; thus, PWV is inappropriate as an index to assess functional stiffness. Recently, the cardio-ankle vascular index (CAVI) was proposed as an index of arterial stiffness in the arterial tree from the origin of the aorta to the ankle. The CAVI is not affected by blood pressure at the measuring time [14,15]. The CAVI reflects functional stiffness and structural stiffness [16,17]. Functional stiffness reflects medial smooth muscle contraction. 

By the way, a predictive sign for impending cardiovascular events is not established yet, although many risk factors were mentioned for the formation of atherosclerosis and vulnerable plaque. It is known that cardiovascular events occur frequently after natural disasters, such as big earthquakes [18] and sporting events [19]. We measured the CAVI of the people living 300 km away from the epicenter of the great East Japan Earthquake in 2011. The CAVI values in healthy people and people with atherosclerotic diseases were transiently enhanced [18]. In addition, we met several people who showed a rapid rise in the CAVI several weeks or months before suffering cardiovascular events. To explain these sequences, we proposed a new hypothesis explaining the processes of plaque rupture: a rapid rise in the CAVI means medial smooth muscle contraction, strangling vasa vasorum, leading to ischemia and necrosis. Finally, plaque rupture occurs [20]. 

In this review, the role of oxidized cholesterol in the formation of atheromatous lesions in the intima, leading to vulnerable plaque, is discussed. Then, the meanings of vasa vasorum rich in atheromatous lesions and penetrating the medial smooth muscle layer from adventitia and a rapid CAVI rise, which reflects arterial smooth muscle contraction, were discussed. Then, the process to propose the hypothesis that medial smooth muscle contraction induces plaque rupture was introduced. Finally, various treatments to improve the CAVI were reviewed. 

## 2. Reconsideration concerning the Formation of Vulnerable Plaque 

The formation of atherosclerotic lesions has been studied by many researchers, and several hypotheses have been proposed [2,3,4,5,6,7]. One important component is believed to be cholesterol, as cholesterol has been found to be the main deposited compound in human atheromatous lesions in the artery [2]. Cholesterol is carried by LDLs, intermediate-density lipoproteins (IDLs), and/or small dense LDLs and is believed to enter the intimal lesion via intimal endothelial cells [3]. There, denatured LDLs are taken up by macrophages, and the macrophages are converted to foam cells via the scavenger pathway, making a lipid pool in the intima [4,5]. Then, an inflammatory reaction occurs, and vulnerable plaque is formed, leading to plaque rupture [7,8]. In this process of progression of atherosclerosis, how the lipid pool was formed adjacent to intimal lamina and how the inflammatory reaction occurred in the lipid pool must be reconsidered to understand the process of plaque rupture. 

### 2.1. The Formation of the Lipid Pool in the Intima of the Arterial Wall 

One first question is the location of the lipid pool in the arterial wall. Human pathological studies showed that cholesterol deposition in the artery is not just observed under the intimal endothelial cell layer [21,22]. The lipid pool is situated deep in the intima, adjacent to the internal lamina. Cholesterol-rich lipoproteins such as LDL, IDL, and/or small dense LDL in the blood have been generally thought to enter through the intimal endothelial layer from the lumen of the artery [23,24]. If the main entrance of cholesterol-rich lipoproteins was the surface of the endothelial layer of the lumen of the artery, the lipid pool would be formed just under the endothelial layer. However, why does the cholesterol pool develop in the deep area adjacent to the internal lamina in the intima and not just beneath the endothelial layer? Recently, several studies have reported that vasa vasorum exist in the arterial wall and become abundant in advanced atherosclerotic lesions [25,26], but the significance of vasa vasorum has not been fully discussed. Nourishment of arteries had been believed mainly by diffusion from the lumen of the vessel. However, recently, it has also been reported that cholesterol-rich lipoproteins and nutrients are transported by vasa vasorum from the adventitia and enter the arterial wall [27] (Figure 1). Vasa vasorum originate at the adventitia, penetrate the medial smooth muscle layer and internal lamina, and then reach the intimal area. If cholesterol-carrying lipoproteins, such as LDL, IDL, and small dense LDL, are carried by vasa vasorum from the adventitia to the intima through the medial smooth muscle layers, it is easily accepted that a lipid pool composed of cholesterol is formed near the internal lamina at the deep intimal layer.

### 2.2. How Does the Lipid Pool Developed in the Intima Make Progress to Vulnerable Plaque—The Role of Oxidized Cholesterol for Provoking Inflammatory Reaction

Deposited cholesterol might not be toxic because cholesterol is a natural organic compound and a part of the cell membrane and is also an ingredient of hormones and bile, as mentioned above. Then, how does the cholesterol pool form and invade the surrounding area? One possible explanation is based on oxidized cholesterols. In atherosclerotic lesions, there are many oxidized cholesterols, such as 7α-ketocholesterol and 7α-hydroxycholesterol [28]. Those oxysterols are generated by various oxidative stresses, such as diabetes mellitus, smoking, and aging during deposition in the lipid pool [29,30]. Furthermore, oxysterols are enzymatically produced [31,32]. The toxicity of oxysterols has been investigated in several studies. 7-ketocholesterol mediated inflammation in atherosclerosis [33,34,35]. Thus, oxysterols might be the main cause to provoke an inflammatory reaction in atheromatous lesions. Furthermore, Ohtsuka et al. reported that deposited lipid containing oxysterol induces apoptosis of the SMC [36] (Figure 2A). Additionally, 7-ketocholeterol induces apoptosis of the SMC [37]. We also observed apoptosis of the SMC in the human coronary artery (Figure 2B). These results suggest that the cholesterol pool might expand by extinguishing the surrounding SMC by inducing apoptosis in the intimal region, rendering the complicated lesion fragile. From these findings, it seems that the risk of cholesterol for arteriosclerosis is not only caused by serum LDL or IDL cholesterol levels but also the number of oxidized cholesterols in the intima. It is likely that those oxidized cholesterols in the lipid pool emerge into circulation. Akiyama et al. reported that the serum oxycholestrol is related to the progression of coronary atherosclerosis [38]. Song et al. reported the association of plasma 7-ketocholesterol with cardiovascular outcomes and total mortality in patients with coronary artery disease [39]. From these observations and phenomena, anti-oxidative therapy, in addition to lowering serum cholesterol levels, must be strictly promoted to prevent the progression of atherosclerosis.

Chronic kidney disease (CKD) is one of the arteriosclerotic diseases, and it is known that oxidative stress influences its progression and complications. However, the role of antioxidants has not been clarified. It is known that probucol, a cholesterol-lowering agent that also lowers HDL cholesterol, has antioxidant activity [40]. We administered probucol to patients with diabetic nephropathy and followed up for 5 years. Probucol administration decreased LDL cholesterol. Furthermore, probucol prevented the progression of CKD and decreased the proportion of patients requiring hemodialysis compared to non-administered patients [41]. Yamashita et al. reported that probucol administration to familiar hypercholesterolemic patients decreases the incidence of cardiovascular events in Japan, despite the accompanying decrease in HDL cholesterol [42]. Once again, the importance of antioxidative therapy for the prevention of arteriosclerotic diseases must be re-emphasized. 

### 2.3. Neovascularization of Vasa Vasorum in the Arterial Wall 

Recent observations have confirmed the presence of vasa vasorum and neocapillaries in atherosclerotic plaques, as mentioned before [25]. This angiogenic process can be initiated by hypoxia in the intima [26]. Hypoxia is induced by the impairment of oxygen diffusion from lumina by intimal thickening; this hypoxic condition induces the expression and release of angiogenic factors (e.g., VEGF) [43]. Thus, the vasa vasorum density is higher in atherosclerotic-prone areas, and neovascularization facilitates the supply of nutrients and oxygen to intimal lesions [27], as illustrated in Figure 1. This neovascularization also augments further lipid deposition, macrophage infiltration, and intimal SMC proliferation. 

The features of vasa vasorum are that it is derived from adventitia, passes through medial smooth muscle layers, and reaches the intimal lesion. As a result, the blood into the intimal layer is carried by vasa vasorum, and its flow volume is under the control of the contractional state of the medial smooth muscle cell (SMC).

As it is not clear how much blood is carried by vasa vasorum to the intimal atheromatous lesion, we observed the surgical endarterectomy of the carotid artery in a patient suffering from carotid artery stenosis. Just after peeling off the proliferative intimal layer of the stenotic carotid artery, the surface of the denuded medial SMC layer was immediately covered with blood, which exuded from the medial smooth muscle layer. This observation means that the blood into the intimal lesion is obviously transported by vasa vasorum penetrating the medial SMC layers from the adventitia (Figure 3A). During this operation, when the surface of the denuded medial smooth muscle layer was covered with a sheet of gauze dipped in saline, the bleeding on the surface of the smooth muscle layer did not stop; however, when this surface was covered with noradrenaline-dipped gauze, the blood exudation stopped (Figure 3B) [20]. This study and observation series revealed that blood in the atheromatous lesion of the intima was transported by vasa vasorum, which penetrated the medial smooth muscle layer. Based on these observations, it is noteworthy that the blood supply into the intimal atheromatous lesion was controlled by the contractional state of the medial smooth muscle layer. 

The next questions are how to guess (1) the degree of the atheromatous lesion and (2) the blood suppling condition into the intimal atheromatous lesion from the adventitia via vasa vasorum by a noninvasive measurement. 

## 3. The Meaning of a Rapid Increase in the CAVI

### 3.1. What Is the CAVI?

To know the stages of arteriosclerosis noninvasively is difficult. Arterial stiffness indicates the degree of arteriosclerosis, and several indices reflecting arterial stiffness have been proposed. PWV reflects arterial stiffness and can be easily calculated by dividing the length of the artery by the time during the pulse spreading from one end to the other end. Various kinds of pulse wave velocity (PWV), such as carotid–femoral PWV (cfPWV [44] and brachial-ankle PWV (baPWV) [45], have been presented in the many papers published in the last 30 years. However, PWV depends on blood pressure at the measuring time [13]. Therefore, PWV is not suitable to measure accurate arterial stiffness at different conditions of blood pressure. Therefore, it is difficult to measure arterial stiffness as vascular function using PWV. 

More recently, the cardio-ankle vascular index (CAVI) was proposed as a new arterial stiffness index of the arterial tree from the origin of the aorta to the ankle [14]. The CAVI was derived from the combination of the stiffness parameter β [46] and the modified Bramwell–Hill equation [47], and the formula is shown below (Equation (1)). The notable feature of the CAVI is its independence from blood pressure at measuring time.
(1)$$CAVI=a\times \left(2\rho \times \frac{\ln(\frac{Ps}{Pd})}{\Delta P}\times {PWV}^{2}\right)+b$$

(PWV, pulse wave velocity of the arterial tree from the origin of the aorta to the ankle; Ps, systolic blood pressure; Pd, diastolic blood pressure; ρ, blood density; and ΔP, Ps–Pd; a, b, coefficients [9].)

The CAVI is measured in the supine position using the VaSela system (Fukuda Denshi, Tokyo, Japan), as illustrated in Figure 4 by Equation (1). The independen from blood pressure is based theoretically on the original stiffness parameter β, which is independent from blood pressure [46]. And this has also been experimentally proven in clinical studies using an α-blocker, doxazosin, and a β-blocker [16]. When the β-blocker metoprolol decreases blood pressure, the CAVI remains changed, whereas, when doxazosin decreases blood pressure, the CAVI also decreases; these results indicate that the CAVI is independent of blood pressure at measuring time and reflects functional stiffness based on the status of SMC contraction.

### 3.2. The CAVI Reflects the Degree of Atherosclerosis

The cut-off value of the CAVI for arteriosclerosis was tentatively defined as nine. The CAVI increases with age, and it is reported to be high in arteriosclerotic patients with coronary artery disease, cerebral infarction, and chronic kidney disease (CKD) [15]. Figure 5 shows pictures of various atherosclerotic stages of the aorta. A CAVI = 7.3 indicates a nearly normal aorta in a 50-year-old woman. A CAVI = 11.0 represents far-advanced stages of atherosclerosis. Furthermore, patients with the most coronary risk factors, such as hypertension, diabetes mellitus, hypercholesterolemia, sleep apnea syndrome, and metabolic syndrome, showed a significantly high CAVI [15]. The above findings indicate that the CAVI reflects the structural stiffness of the arterial tree.

### 3.3. The CAVI Also Reflects the Functional Stiffness of the Artery

An interesting feature of the CAVI is that it reflects functional stiffness, which is derived from the contractional state of arterial smooth muscles. This conclusion can be deduced by the following studies. The administration of the adrenaline receptor α-blocker, doxazosin, decreases the CAVI, as stated above [16]. Nitroglycerin administration also decreases the CAVI among control subjects and patients with arteriosclerotic diseases [17]. The latter evidence indicates that the medial smooth muscle layer, even in advanced stages of atherosclerosis, retains the ability to contract or dilate in response to various stimuli. Septic conditions decrease the CAVI, accompanied by a decrease in blood pressure [48]. Miyazaki et al. reported that enhanced intracranial pressure provokes the enhancement of blood pressure and the CAVI in rabbits [49]. Taken together, those results indicate that the CAVI reflects contraction or dilation of arterial smooth muscle; a rapid change in the CAVI indicates arterial smooth muscle contraction.

### 3.4. The CAVI Just after a Huge Natural Disaster and Stress

There were several papers reporting that many more cardiovascular events occurred immediately after huge disasters, sometimes accompanying high blood pressure [50]. Trichopoulos et al. reported that psychological stress-induced fatal heart attacks at the time of the earthquake [51]. Dobson et al. reported that the number of heart attacks increased after the Newcastle earthquake [52]. Leor et al. also reported that sudden cardiac death was triggered by an earthquake [53]. Regarding the cause of those earthquake-related sequences of cardiovascular events, psychological stress was mentioned; however, the precise mechanism was not fully evaluated. The huge earthquake “Great East Japan earthquake” occurred in the northeastern part of Japan in 2011. At our research center, Toho University Sakura Medical Center, in Chiba, located 300 km from the epicenter, we measured the CAVI of healthy colleagues working in our hospital just after the earthquake and 2 and 4 weeks later. Compared with the value just after the earthquake, the CAVI decreased after 2 and 4 weeks. Moreover, patients with arteriosclerotic diseases who came to our hospital periodically showed a transient increase in the CAVI at 1 week after the earthquake and a decrease after a few months [18]. In the several days following the earthquake, the number of patients suffering from brain hemorrhage who visited our hospital increased by two-fold compared with before the earthquake. The number of deaths in Sakura City during the few months following the earthquake was 30% higher compared with the same time frame in the years leading up to the disaster [18]. The severe disaster provoked cerebrovascular events and death, accompanied by a rapid rise in the CAVI.

Furthermore, we encountered a patient whose CAVI rapidly increased and then suffered from a brain hemorrhage 2 weeks later. Another patient showed an increased CAVI and then suffered from myocardial infarction 4 months later. One patient suffered from an aortic dissecting aneurysm one month after a rapid rise in the CAVI. These cases might be just incidental; however, those series of events may point out the importance of a rapid rise in the CAVI as a prodrome of cardiovascular events. Large prospective studies are needed to confirm the relationship between a rapid rise in the CAVI and subsequent cardio-cerebrovascular events. However, if the relationship between cardiovascular events and the rapid rise in the CAVI is confirmed, the following hypothesis might be proposed.

### 3.5. Smooth Muscle Cell Contraction Hypothesis for Plaque Rupture: The Role of Rapid Rise in the CAVI

The proposed mechanism for the occurrence of cardiovascular events after the rapid rise in the CAVI in cases showing a high CAVI is as follows. At first, an atheromatous lesion is formed by infiltration of cholesterol-rich lipoproteins such as LDL, IDL, and small dense LDL in a lipid pool. Then, deposited cholesterols become oxidative products due to oxidative stress. The resultant oxysterols act as a toxic substance in the intima, which provokes inflammatory reactions, such as the stimulation of macrophages, promoting the migration and proliferation of SMCs and inducting apoptosis of SMCs, eventually leading to the expansion of the lipid pool. Furthermore, the inflammatory reaction and anoxia in the intima thickening lesion stimulate neovascularization. Then, the vasa vasorum became developed. At this stage, the CAVI gradually increases. A high CAVI means the presence of vulnerable plaque in the arterial tree and coronary artery. These processes of arteriosclerotic lesion formation (Stage I) are illustrated in Figure 1, and subsequent SMC contraction and plaque rupture (Stage II) is illustrated in Figure 6.

When a huge natural disaster, such as a severe earthquake or major psychological stress occurs for people with a high CAVI, a rapid rise in the CAVI is observed. This means that medial SMCs contract. This contraction of the SMC strangles the vasa vasorum, which penetrate the medial SMC layer; consequently, the blood supply to the intimal atheromatous lesions stops. This process leads to vulnerable plaque ischemia and necrosis. Subsequently, the rupture of vulnerable plaque occurs. Considering brain arteries in those with a high CAVI, a rapid rise in the CAVI might mean a disruption of the blood supply to brain arteries, leading to necrosis of brain arteries, following brain hemorrhage. In the case of coronary arteries in those with a high CAVI, a rapid rise in the CAVI means a lack of blood supply to vulnerable plaque. Plaque rupture causes thrombus formation in the lumen of the artery, leading to myocardial infarction. In the case of the aorta, a rapid rise in the CAVI causes the strangulation of vasa vasorum and cessation of the blood supply to the intima, leading to necrosis in the intimal layer. Intimal necrosis renders dissection of the aorta. Osada et al. reported that aortic dissection in the outer third of the media is related to the significance of vasa vasorum in the triggering process [54]. If the authors dare to state tentative values, a basal value in the CAVI > 10 may expect to have advanced stages of arteriosclerosis with vulnerable plaque, which is close to the mean CAVI value (7.84) in Japanese + 2 × standard deviation (2 × 1.07). An enhanced ΔCAVI of >0.7 would correspond to a two-fold increase in the variation coefficient (3.7%) of the CAVI measurement [14]. Thus, we proposed that a rapid rise in the CAVI (ΔCAVI > 0.7) in those with a basal CAVI (>10) might be a prodrome of serious cardiovascular events in the near future. Based on this hypothesis, the duration of the rapid rise in the CAVI becomes important for plaque rupture. The CAVI enhanced during a huge earthquake and decreased two weeks later (18), and a decreased CAVI was observed a half year later, indicating that the duration of rapid rise in the CAVI, inducing plaque rupture, may be transient. Of course, its duration and degree of causing plaque rupture is dependent on the basic condition of the vulnerability of the plaque. Actually, further studies on the continuous monitoring of the CAVI among massive cases will be needed in the future in order to clarify a more precise causal relationship between the rapid rise in the CAVI and event occurrence.

Anyway, based on the “SMC contraction hypothesis of plaque rupture” [20], monitoring the CAVI frequently in daily life might be useful to elicit caution toward life-threatening cardiovascular events in the near future.

## 4. Treatments to Improve the CAVI

The aforementioned sequences of events occurring in the arteriosclerotic lesion might support the “SMC contraction hypothesis for plaque rupture”. The next question is how to prevent a rapid rise in the CAVI, which induces plaque rupture in advanced stages. To date, several treatments or methods to improve the CAVI have been reported (refer to [15]). Among them, the main treatments are summarized in Table 1.

(1)Lifestyle changes

Weight reduction decreases the CAVI in obese diabetic patients. Continuous positive airway pressure therapy for sleep apnea syndrome decreases the CAVI (refer to 15). Waon therapy, in which patients are placed in a dry sauna at 60 degrees for 15 min, improves the CAVI [55].

(2)Controlling high blood pressure

Antihypertensives, such as angiotensin-receptor blockers (olmesartan) and calcium channel blockers (cilnidipine and efonidipine), are reported to decrease the CAVI (refer to [15]).

(3)Control of diabetic mellitus

The response of the CAVI to glucose-lowering treatment depends on the type of agent. Pioglitazone, a rapid-acting insulin, dipeptidyl peptidase 4 inhibitors, anagliptin, and the new sulfonylurea, glimepiride, are reported to decrease the CAVI (refer to [15]).

(4)Among lipid-lowering agents

Among the many available statins, pitavastatin decreases the CAVI. Bezafibrate and eicosapentanoic acid are also reported to decrease the CAVI (refer to [15]).

(5)Health supplements

For supplements, sirtuin enhancers, resveratrol, and s-equal are reported to decrease the CAVI in a double-blind study. The CAVI might be a useful index to evaluate the effects of supplements (refer to [15]).

Further studies are necessary to confirm that a decreased CAVI improves the risk of mortality and morbidity.

## 5. Conclusions and Future Directions

To prevent the progression of atherosclerosis, antioxidant therapy should be re-emphasized. The role of vasa vasorum in supplying the blood to arteriosclerotic lesions and medial smooth muscle contraction in the progression of vulnerable plaque should be given greater attention. As an index reflecting medial smooth muscle contraction, a rapid rise in the CAVI might be a warning sign for impending cardiovascular events.

In order to forecast impending cardiovascular events, periodic monitoring of the CAVI is needed.

To prevent cardiovascular events, a rapid rise in the CAVI should be treated with various treatments, especially relieving mental stress. To strengthen this hypothesis, large prospective studies are required.

## Figures and Tables

**Figure 1 jcm-12-07436-f001:**
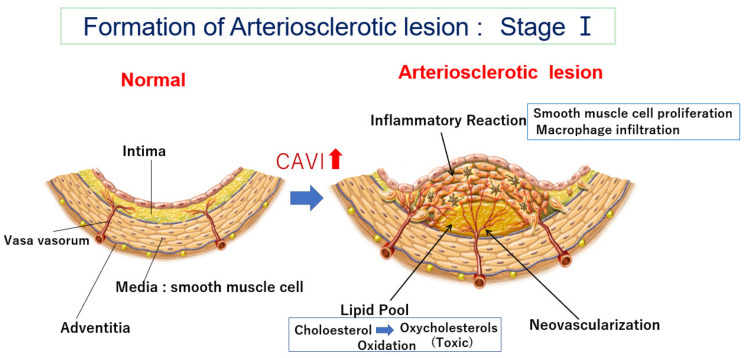
Formation of an arteriosclerotic lesion: Stage Ⅰ. Cholesterol-rich lipoproteins such as LDL, IDL, and small dense LDL are carried by vasa vasorum and enter the deep intimal area. There, cholesterols are oxidized, and several oxidative products are produced. Oxysterols are toxic and damage the surrounding tissues. For example, 7-ketocholeterol induces apoptosis of smooth muscle cells and expands the lipid pool. Then, oxysterols induced an inflammatory reaction, and macrophages infiltrated. Smooth muscle cells migrate from the media to the intima and proliferate to make intimal thickening. During this process, vasa vasorum develop from adventitia into an intimal lesion through the medial smooth muscle layer (neovascularization). The CAVI increases as arteriosclerosis develops. CAVI: cardio-ankle vascular index; IDL: intermediate-density lipoprotein; LDL: low-density lipoprotein.

**Figure 2 jcm-12-07436-f002:**
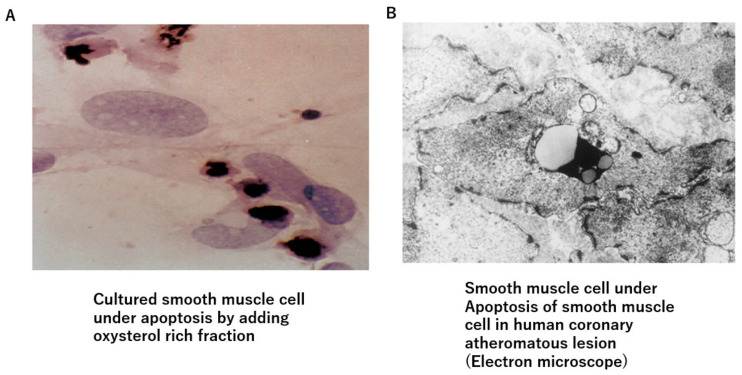
Oxysterol-induced apoptosis of smooth muscle cells. (**A**) Oxysterol-rich fraction-induced apoptosis of cultured smooth muscle cells. (**B**) Smooth muscle cells are observed to fall into apoptosis in human arteriosclerotic lesions.

**Figure 3 jcm-12-07436-f003:**
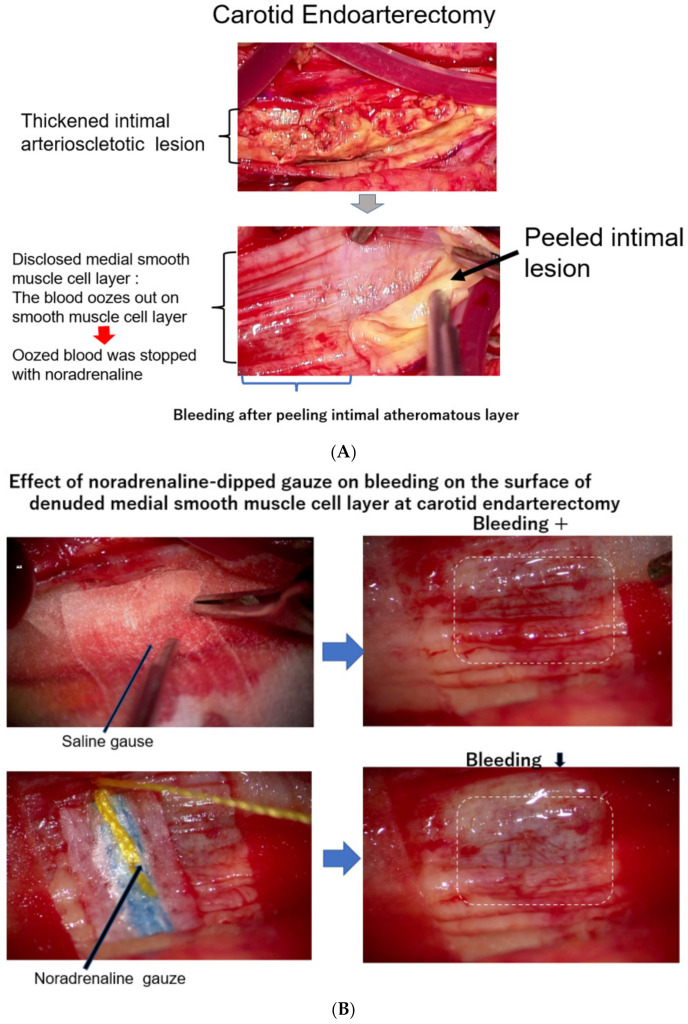
(**A**) The surface of the medial layer peeled off the intimal atheromatous layer. At endarterectomy of the carotid artery, the intimal atheromatous layer could be peeled away from the medial smooth muscle cell layer. The denuded surface of the medial smooth muscle layer was promptly covered with blood, indicating that the intimal arteriosclerotic lesion was supplied by blood with vasa vasorum, which penetrated through the medial smooth muscle layer from the adventitia. (**B**) Bleeding at the surface of the peeled medial layer of the arteriosclerotic artery at endarterectomy. When the surface of the smooth muscle cell layers was covered with a sheet of gauze dipped with norepinephrine, the blood exuding from the medial smooth muscle layer was stopped, indicating that contraction of the medial smooth muscle stopped blood supply from the adventitia and caused ischemia of the intimal lesion. The medial smooth muscle contraction would bring a rapid increase in the CAVI.

**Figure 4 jcm-12-07436-f004:**
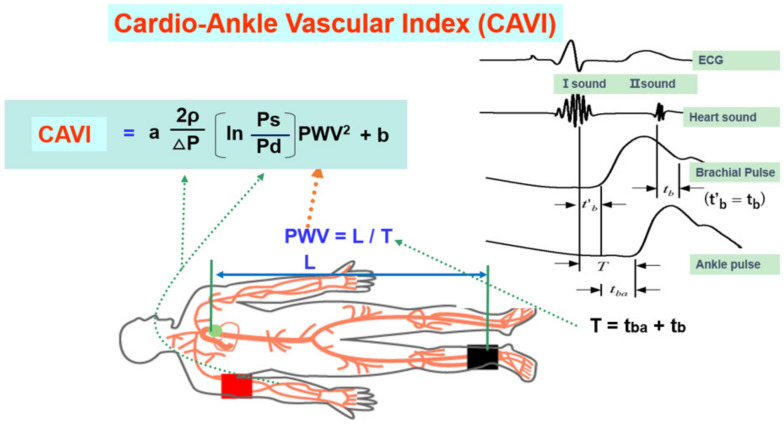
Equation of the cardio-ankle vascular index and measuring methods (adapted from Ref. [14]). Ps: systolic blood pressure, Pd: diastolic pressure, In: natural logarithm, ΔP: Ps-Pd, ρ: blood viscosity, a,b: coefficiency, PWV: pulse wave velocity, L: length from the origin of the aorta to the ankle, T: time taken for the pulse wave to propagate from the aortic valve to the ankle, tba: time between the rise of brachial pulse wave and the rise of ankle pulse wave, tb: time between aortic valve closing sound and the notch of brachial pulse wave, t’b: time between aortic valve opening sound and the rise of brachial pulse wave.

**Figure 5 jcm-12-07436-f005:**
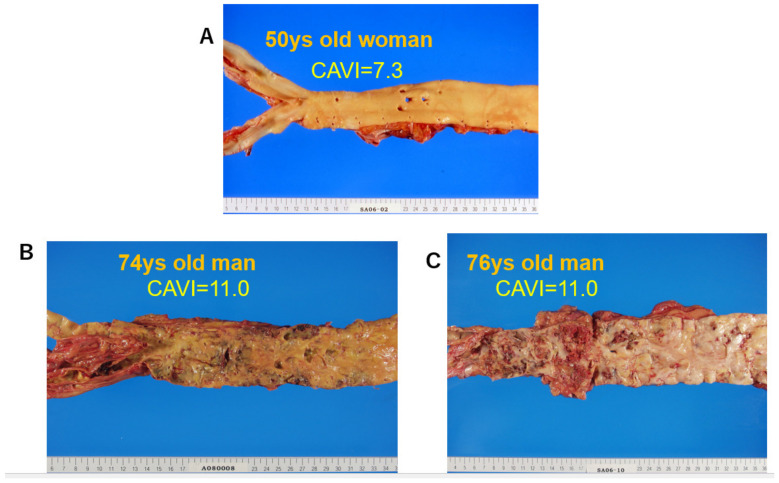
Various stages of atherosclerosis of the aortae and the CAVI. (**A**) Aorta of a 50-year-old woman. The CAVI was 7.3, which is a nearly normal level (−0.5SD). (**B**) Aorta of a 74-year-old man. The CAVI was 11.0, which is high for his age (+2SD). (**C**) Aorta of a 76-year-old man. The CAVI was 11.0, which is high for his age (+2SD). A high CAVI value over +2SD from the average value for ages might indicate the presence of arteriosclerosis with vulnerable plaque. CAVI: cardio-ankle vascular index.

**Figure 6 jcm-12-07436-f006:**
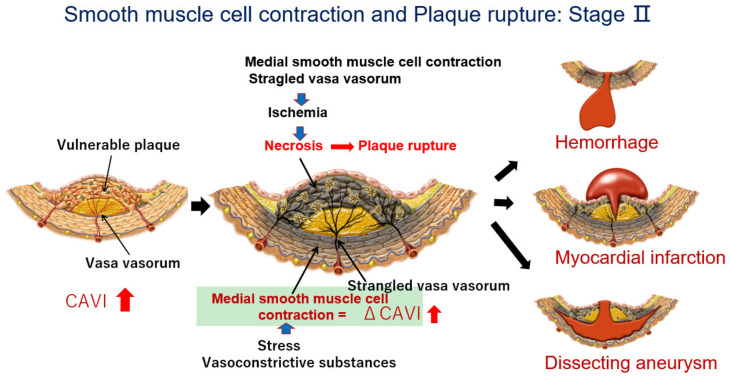
The process of plaque rupture of vulnerable plaque triggered by medial smooth muscle contraction monitored with a rapid rise in the CAVI: Stage II. When a rapid rise in the CAVI was observed, medial smooth muscle contraction occurred. Then, the vasa vasorum are strangled, and blood supply to the intimal lesion is stopped, which causes ischemia and necrosis of vulnerable plaque, following plaque rupture. Cardiovascular events such as cerebral bleeding, myocardial infarction, and dissecting aneurysms in the aorta might happen in the near future (Quoted from Ref. [20]).

**Table 1 jcm-12-07436-t001:** Treatments and Medicines improving High CAVI.

Life Style Change	Antihypertensives	Anti-Diabetic Drugs	Lipid Lowering Agents
Weight reduction	ARB: Olmesartan	Rapid acting insulin	Pitavastatin
Visceral fat reduction	CCB: Cilnidipine	Pioglitazone	Bezafibrate
CPAP therapy for SAS	Efonidipine	Glimepirid	Eicosapentaenoic acid
Exercise			
Smoking cessation	Supplements		
Waon therapy	Resveratrol		
	s-Equal		

ARB, angiotensin receptor blocker; CCB, calcium channel blocker; CPAP, continuous positive airway pressure; SAS, sleep apnea syndrome.

## Data Availability

No new data were created or generated in this manuscript.

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
