# Peer review of "The Process of Plaque Rupture: The Role of Vasa Vasorum and Medial Smooth Muscle Contraction Monitored by the Cardio-Ankle Vascular Index"

_jcm, 2023, doi:10.3390/jcm12237436_

Round 1
Reviewer 1 Report
Comments and Suggestions for Authors
The manuscript is of potential interest and overall well written. However, some aspects should be addressed:
#1. In the abstract, lines 27-29, it appears that this an original article. However, in the section Introduction, line 82, it is clearly stated that this is a review article. Abstract should reworded accordingly, Else, clarify the nature of the manuscript.
#2. On page 6, authors should provide additional information on how CAVI is measured. I would appreciate if they provided the equation to calculate CAVI. In addition, they should clarify how CAVI is measured (seated or sitting position), which device should be used and if this has been validated or not, and if some specific necessities are needed.
#3. Authors should clarify if the “rapid rise” of CAVI, as described as a potential cause of acute cardiovascular events, is a transient or permanent phenomenon or not. After earthquake it is reasonable to have rapid, but time-limited, blood pressure elevations, that may resolved in the subsequent hours/days. Please clarify when CAVI has been assessed before earthquake, if available, and if it remained elevated or not.
Minor points:
- Some abbreviations (e.g. baPWV or cfPWV) should be explained.
- On page 7, lines 265-266 larger characters are adopted apparently without reason.
- In figure 5, stage 2 is described, but I cannot see stage 1 summary.
Comments on the Quality of English LanguageMinor English editing is required to improve the reading of the manuscript.
Author Response
Thank you for your review of our paper. All comments were reasonable and helpful to understand the meaning this paper more accurately. So, we corrected or added the explanations to each comment as follow.
#1. In the abstract, lines 27-29, it appears that this an original article. However, in the section Introduction, line 82, it is clearly stated that this is a review article. Abstract should reworded accordingly, Else, clarify the nature of the manuscript.
We agreed. So, we changed as follow:
In the abstract, lines 27-29.
We reported ――――It is reported
We observed------------There are
#2. On page 6, authors should provide additional information on how CAVI is measured. I would appreciate if they provided the equation to calculate CAVI. In addition, they should clarify how CAVI is measured (seated or sitting position), which device should be used and if this has been validated or not, and if some specific necessities are needed.
We understood. It is good advice. We added the equation, measuring machine and
measuring schema( Fig 4).
Cardio-ankle vascular index (CAVI) reflects the arterial stiffness of the arterial tree from the origin of the aorta to the ankle. CAVI is derived from stiffness parameter β( ), and Bramwell-Hill’s equation( ), and is obtained by systolic and diastolic blood pressures and pulse wave velocity (). CAVI is measured in supine position using VaSela system (Fukuda Denshi, Tokyo、Japan) as illustrated by Fig 4 by eq.18).
eq.1
(PWV, pulse wave velocity of the arterial tree from the origin of the aorta to the ankle; Ps, systolic blood pressure; Pd, diastolic blood pressure; ρ, blood density; , Ps–Pd; a, b, coefficients [9]).
#3. Authors should clarify if the “rapid rise” of CAVI, as described as a potential cause of acute cardiovascular events, is a transient or permanent phenomenon or not. After earthquake it is reasonable to have rapid, but time-limited, blood pressure elevations, that may resolved in the subsequent hours/days. Please clarify when CAVI has been assessed before earthquake, if available, and if it remained elevated or not.
- Definition of Rapid rise of CAVI:
Rapid rise of CAVI reflects arterial smooth muscle contraction, then, it is mostly transient. However, it is difficult to differentiate rapid rise or permanent rise in a moment. At huge earthquake, CAVI of healthy people just at the earthquake decreased 2 weeks later. Those of arteriosclerotic persons, enhanced CAVI comparing 6 months before, decreased 6 months later( Ref) Anyway, gradual permanent rise means the progression of arteriosclerosis. Transient rise ( over 0.7, a few weeks to a few months ) rise, might be dangerous in persons with vulnerable plaque.
Ref: 18 Kazuhiro Shimizu 1, Mao Takahashi, Kohji Shirai J Atheroscler Thromb 2013;20(5):503-11. doi: 10.5551/jat.16097. Epub 2013 Mar 25. A huge earthquake hardened arterial stiffness monitored with cardio-ankle vascular index
Then, those were summarized and described in -----------.P9, line34
Based on this hypothesis, the duration of rapid rise of CAVI becomes important for plaque rupture. CAVI enhanced at huge earthquake and decreased two weeks later (18), and decreased CAVI was observed a half years later, indicating that the duration of rapid rise of CAVI inducing plaque rupture may be transient. Of cause, its duration and degree to cause plaque rupture is dependent on the basic condition of vulnerability of the plaque. Actually, further studies on continuous monitoring CAVI among massive cases will be needed in the future in order to clarify more precise causal relationship between rapid rise of CAVI and event occurrence.
Anyway, based on the “SMC contraction hypothesis of plaque rupture” [20], monitoring of CAVI frequently in daily life might be useful to elicit a caution of the life-threatening cardiovascular events in near future
Minor points:
- Some abbreviations (e.g. baPWV or cfPWV) should be explained.
baPWV is brachial ankle pulse wave velocity and cfPWV is carotid femoral pulse wave velociry. Those were added in the text.
- On page 7, lines 265-266 larger characters are adopted apparently without reason.
Larger characters were adjusted
- In figure 5, stage 2 is described, but I cannot see stage 1 summary.
We changed the summary of Fig 1: stage 1
Reviewer 2 Report
Comments and Suggestions for Authors
Summary
.
This review manuscript (article) addresses the issue regarding atherosclerotic plaque rupture. Finally, the authors claim that a rapid rise of high cardio- ankle vascular index (CAVI) may predict rupture of vulnerable plaque and might be a warning sign for impeding cardiovascular events.
I enjoyed to read this manuscript and thanks to the authors who described possible mechanism of vulnerable plaque rupture.
Evaluation
My comments are as follows.
(1) Abstract: Like in an original research article, the authors claimed the usefulness of CAVI in this review article. They should rewrite the description into the abstract for review article. Please omit the sentences from line 7, described “We reported rapid rise -----in peoples after big earthquakes,------.)
(2) Introduction: p 2, line 5, misspell (rumen: lumen). Page 2, paragraph 4, misspell (earth quake: earthquakes)
(3) Please describe the name of the equipment measuring CAVI and also the name of company. When the authors have conflict of interest (COI), please disclose the COI.
Comments on the Quality of English Language
NA
Author Response
Summary.
This review manuscript (article) addresses the issue regarding atherosclerotic plaque rupture. Finally, the authors claim that a rapid rise of high cardio- ankle vascular index (CAVI) may predict rupture of vulnerable plaque and might be a warning sign for impeding cardiovascular events.
I enjoyed to read this manuscript and thanks to the authors who described possible mechanism of vulnerable plaque rupture.
Thank you very , very much. We appreciates your understanding our intention.
Evaluation
My comments are as follows.
- Abstract: Like in an original research article, the authors claimed the usefulness of CAVI in this review article. They should rewrite the description into the abstract for review article. Please omit the sentences from line 7, described “We reported rapid rise -----in peoples after big earthquakes,------.)
We changed as follows
We reported ――――It is reported
We observed------------There are
- Introduction: p 2, line 5, misspell (rumen: lumen). Page 2, paragraph 4, misspell (earth quake: earthquakes)
We corrected: rumen – lumen
Earth quake – earthquake
- Please describe the name of the equipment measuring CAVI and also the name of company. When the authors have conflict of interest (COI), please disclose the COI.
We added he name of the equipment measuring CAVI and also the name of company.
We stated “the authors have no conflict of interest (COI)” in the last.
Comments on the Quality of English Language NA
Submission Date
01 October 2023
Date of this review
17 Oct 2023 07:51:59
Round 2
Reviewer 2 Report
Comments and Suggestions for Authors
The revised manuscript has been improved.